**Data Availability Statement:** All relevant data are within the manuscript and its Supporting Information files.

# Barriers and facilitators to pharmacovigilance activities in Pakistan: A healthcare professionals-based survey

Rabia Hussain👤[1]*, Tayyaba Akram[2], Mohamed Azmi Hassali[1], Jaya Muneswarao[3], Anees ur Rehman[4], Furqan Hashmi[5], Zaheer-Ud-Din Babar[6]

1 Department of Social and Administrative Pharmacy, School of Pharmaceutical Sciences, Universiti Sains Malaysia, Pulau Pinang, Malaysia, 2 Department of Mathematics, COMSATS Institute of Information Technology, Lahore, Pakistan, 3 Hospital Pulau Penang, Jalan Residensi, George Town, Pulau Pinang, Malaysia, 4 Department of Pharmacy Practice, Faculty of Pharmacy, Bahauddin Zakariya University, Multan, Pakistan, 5 University College Pharmacy, The University of Punjab, Lahore, Pakistan, 6 Department of Pharmacy, University of Huddersfield, Queensgate, Huddersfield, United Kingdom

* rabia.hussain2010@gmail.com

## Abstract

The timely reporting of adverse drug reactions (ADRs) could improve pharmacovigilance (PV) in a healthcare system. However, in almost all healthcare systems barriers exist that lead to the underreporting of ADRs. The objective of this study was to identify the barriers and facilitators regarding PV activities from the point of view of healthcare professionals (HCPs) in Lahore, Pakistan. A cross-sectional questionnaire-based survey was conducted between September 2018 to January 2019. The data was collected through convenience sampling of physicians, pharmacists, and nurses at tertiary care public hospitals in Lahore. A total of 384 questionnaires were distributed, and 346 HCPs responded to the survey. Over 62% percent of physicians and 54.8% of nurses agreed that they did not know how to report an ADR in their workplace. About 43.2% of pharmacists and 40.1% of nurses disagreed that they were not aware of the need for ADR reporting. Furthermore, 41.6% of nurses identified a lack of financial reimbursement and 51.8% highlighted a lack of support from a colleague as a reason that could lead to the underreporting of ADR. The majority of participants, including 69.6% physicians, 48.6% pharmacists, and 55.3% nurses identified the lack of knowledge about the existence of a national PV centre. Extra time for ADR reporting, incentives, continuous medical education, reminders, and availability of an online ADR reporting system was classed as the facilitators and were agreed upon by the majority of HCPs.

## Introduction

Patients from around the world are continuously under threat from the onset of adverse drug reactions (ADRs), which not only risks their well-being but also poses a great challenge to the healthcare systems in the form of an increased financial burden and workforce requirement [1]. An ADR is defined as a noxious, unintended injury that arises from drug-related use,

**Funding:** The author(s) received no specific funding for this work.

**Competing interests:** The authors have declared that no competing interests exist.

accountable for 6% of hospital admissions and 2% ADRs related mortality [2, 3]. Among hospitalized patients, ADRs represent the fifth most common cause of death [4]. A European Commission-based survey has estimated that 5% of the hospital admissions in Europe were due to the ADRs of the drugs [5]. Moreover, in European Union, about 197,000 deaths are caused by the ADRs, owing to a total cost of €79 billion [6]. In the United States, ADRs are reported to lead as the sixth highest cause of mortality [7]. A hospital-based study from the United Kingdom (UK) has also shown that 15% of UK patients experience ADRs during their admissions [8].

A pharmacovigilance (PV) system is responsible for the detection, assessment, understanding, and prevention of ADRs [9, 10]. Usually, drug-related ADRs are reported to the PV system through various channels in a healthcare system which may include spontaneous ADR reporting systems and post-marketing surveillance, etc. ADRs data, are sometimes incomplete, and unrepresentative, especially the post-hospital admissions-based data are poor [11]. This is mainly associated with the practices of PV in any healthcare system. However, literature has revealed that ADR reporting improved wherever there was an involvement of educational and training-based interventions including enhanced motivational strategies for reporting and enhanced availability of the reporting tools for the ADR reporting staff [12].

The Programme for International Drug Monitoring (PIDM) was first introduced in 1968 and initially, ten countries participated including Australia, Canada, Denmark, Ireland, Germany, Netherlands, New Zealand, Sweden, United States, and the United Kingdom [13]. To date, there are a total of 171 member countries that have joined PIDM, while 145 are full members and 26 are associate member countries [14]. Pakistan joined the PIDM by the World Health Organization (WHO) as the 134th member in 2018, however, PV system is still in its infancy in the country [15]. The PV programme in Pakistan aims to detect ADR signals. The other aims of this programme are to make ADR reporting a mandatory task to be performed by healthcare professionals (HCPs), to start active surveillance, and to expand the scope of PV to all public hospitals including the basic healthcare centres in the country [9].

The medicines safety issue has a wider impact on a country as big as Pakistan with a population of over 200 million people [16]. The incident at the Punjab Institute of Cardiology (PIC) in Lahore, which accounted for the life of over a hundred patients, highlighted the drug safety monitoring issues in the country including negligence on the part of HCPs [17]. Post the PIC incident, the Drug Regulatory Authority of Pakistan (DRAP) developed Pakistan's national PV system. Under the guidelines of DRAP, each healthcare professional, patient, or even caregiver can report an ADR, which is being provided through an online submission system known as MED Vigilance E-Reporting System [18]. These reports are then analyzed and further communicated to the Uppsala PIDM for signal detection [14]. However, since the establishment of DRAP, a total of 6587 ADR reports have been received by the national PV centre. Out of which, only 124 reports had been made by the PV centre in Punjab, however, none of the reports was made by the public [19]. To date, the ADR reporting rate in Pakistan is suboptimal and does not meet the WHO standard for ADR reporting, which is 200 ADR reports per million inhabitants in a year [20]. Therefore, the underreporting of ADR remains a major concern in almost all parts of the country.

Healthcare professionals play an important role in the establishment and functionality of PV in a country. Hence it is important to understand the barriers and facilitators, that could otherwise have a significant impact on HCPs' beliefs, and practices and can help to improve the medicines safety situation in the country [10]. Studies from Pakistan have summarized that HCPs had poor knowledge about PV and ADR reporting and identified a lack of awareness about PV guidelines and the absence of a PV centre as barriers to ADR reporting. As Iffat et al. (2014), Nisa et al. (2018) and Syed et al. (2018) indicated that HCPs had poor knowledge

about ADR reporting systems and the unavailability of incentives, and a professional environment hindered the ADR reporting [21–23]. Previous studies have documented that awareness about the PV system, education and training can improve the ADR reporting among HCPs [23–27]. A recent study has analyzed the core indicators responsible for a functional PV system in Pakistan [19]. Though few studies have highlighted the barriers and facilitators affecting the ADR reporting system among physicians and pharmacists in Pakistan, there is a scarcity of literature about barriers causing underreporting of ADRs and facilitators to improve the reporting of ADRs among HCPs in the country. In this context, this is the first study being conducted to determine the barriers and facilitators related to ADR reporting from all three cadres of HCPs including physicians, pharmacists, and nurses in Lahore, Pakistan.

## Methods

### Study design and study period

A cross-sectional study was conducted to assess the barriers and facilitators among HCPs regarding PV activities in Lahore. The study was conducted between September 2018 to January 2019.

### Study population & study site

All HCPs (including pharmacists, doctors, and nurses) working in the government tertiary care public hospitals were considered eligible to participate in the study. This was considered if they were registered with the relevant provincial/national council and willing to provide written consent to the researcher. Those who refused or were not willing to participate in the study were excluded. The selected tertiary care public hospitals were the representative hospitals of the Punjab and almost served patients from all areas of the province and were well equipped with modern facilities.

### Sample size determination and sampling

The study was conducted through convenience sampling technique. The physicians, pharmacists, and nurses, whereby participants were selected according to their accessibility, convenience, and proximity.

The sample size was calculated to be 384 by employing the Cochrane formula [28].

### Development of survey questionnaire

The questionnaire was developed based on the literature and exploratory interviews-based findings from physicians, pharmacists, and nurses during the first phase of the study [10, 17, 29, 30]. The questions included demographic details and experiences of HCPs regarding barriers and facilitators toward ADR reporting in their work setting.

Section one included demographic characteristics including gender, age, occupation, education, etc. Section two covered barriers related to the ADR faced by HCPs while reporting an ADR. Section three covered areas related to facilitators improving ADR reporting-related activities.

### Validation of survey questionnaire

The questionnaire was tested for its face and content validity and two academics from the School of Pharmaceutical Sciences, Universiti Sains Malaysia (USM) reviewed the questionnaire in terms of its clarity and relevance and was modified according to the suggestions provided. Before the actual survey implementation, the questionnaire was also pilot tested on 30

HCPs. These HCPs were later excluded from the study. The internal consistency of the questionnaire was found to be 0.72.

### Data collection

A face-to-face survey was administered using questionnaires along with the explanatory statement about the research project and consent form in a hard copy.

Questionnaires were provided with a written informed consent form which assured the participants about the confidentiality and anonymity of the gathered information. The self-administered questionnaires were distributed to HCPs after obtaining permission from the relevant heads of the department, nursing heads, and chief pharmacists. The majority HCPs filled and returned the questionnaires on the same day while the remaining questionnaires were collected during periodic visits to the participating hospitals/ departments and reminders were also sent after a week, if deemed necessary. Extra copies of questionnaires were provided to avoid any inconvenience due to shortage/loss of questionnaires and no incentives were given to fill the questionnaires.

### Ethical consideration

The ethical approval was obtained from Humans Ethics Committee (HEC), University College of Pharmacy, University of the Punjab, Lahore, Pakistan with reference no. HEC/PUCP/1943. A brief explanation of the objective of the study was also provided along with the survey questionnaire. This was done to avoid ambiguity and misunderstanding about the survey and the process of data collection.

### Data quality control and analysis

The data were checked for completeness, accuracy, and consistency, and those found incomplete or missing in addressing important variables were discarded. The data were coded with a sequential number and entered into SPSS version 25.0 for analysis. Shapiro-Wilk and Kolmogorov-Smirnov tests were employed to assess the normality of the data. Percentages and frequencies were used to express categorical variables while Mann-Whitney-U and Kruskal-Wallis tests were applied to continuous data and $p<0.05$ was considered significant. For barriers and facilitators related to the underreporting of ADRs, median scores were calculated, and their association was measured with demographic variables.

## Results

### Demographics data

Based on sample size calculation, a total of 384 questionnaires were distributed to HCPs (considering the inclusion criteria). The returned questionnaires were collected by the researcher with a response rate of 90.1% (n = 346) for all participating HCPs. The detailed demographics of the participants are described in Table 1.

### Barriers related to ADR reporting

Broadly, the barriers to ADR reporting can be classified into two categories, comprising of healthcare system-related barriers and individual-related barriers as given in Fig 1 [31].

The barriers related to ADR reporting were evaluated on a three-point Likert scale and responses were presented in Fig 2 for physicians, pharmacists, and nurses respectively. The majority of the respondents strongly agreed to the different aspects of the barriers to ADRs, such as 86 (76.8%) physicians, 28 (75.6%) pharmacists and 142 (72.0%) nurses identified a lack

**Table 1. Demographics of the participants (n = 346).**

| Category | Subcategory | Frequency (Percentage) n (%) |
|---|---|---|
| Healthcare professional type | Physicians | 112 (32.4) |
| | Pharmacists | 37 (10.7) |
| | Nurses | 197 (56.9) |
| Gender | Male | 59 (17.1) |
| | Female | 287 (82.9) |
| Age (Years) | 25–30 | 57 (16.5) |
| | 31–35 | 197 (56.9) |
| | 36–40 | 72 (20.8) |
| | 41–45 | 16 (4.6) |
| | 45 or above | 4 (1.2) |
| Qualification | Bachelors | 245 (70.8) |
| | Masters | 33 (9.5) |
| | Specialization | 68 (19.7) |
| Experience | 5 years or below | 178 (51.4) |
| | 6–10 years | 118 (34.1) |
| | 11–15 years | 26 (7.5) |
| | 16–20 years | 13 (3.8) |
| | 20 or above | 11 (3.2) |

of knowledge about ADRs. Similarly, 68 (60.8%) physicians, 21 (56.7%) pharmacists, and 119 (60.4%) nurses agreed on lack of time is a hindrance to ADR reporting. Among all, the majority of nurses identified a lack of interest to report an ADR and a lack of confidence in discussing an ADR with other colleagues as the reasons for not reporting an ADR. This has also been shown in Table 2 and graphically represented in Fig 2.

Most participants disagreed that the ADR form is too difficult to fill, however, 70 (62.5%) physicians and nurses 108 (54.8%) agreed, that they did not know, how to report an ADR in their workplace and 17 (45.9%) pharmacists and 74 (37.5%) nurses considered fear of legal liability as a barrier.

About one-third of the physicians 47 (41.9%) agreed that they were unaware of the need to report an ADR. Also, the majority of nurses have identified a lack of financial reimbursement 82 (41.6%) and lack of support from colleagues and administration 102 (51.8%) as possible barriers to ADR reporting. The participants including 78 (69.6%) physicians, 18 (48.6%) pharmacists, and 109 (55.3%) nurses identified the lack of awareness about the existence of the national PV centre as a barrier to ADR reporting.

Regarding facilitators to ADR reporting, the majority of the nurses disagreed with certain facilitating factors for ADR reporting such as extra time and incentives but agreed that continuous medical education, reminders, and an online system for reporting as agreed by a majority of the physicians and pharmacists as shown in Table 3, Fig 3.

**Analysis of ADR reporting related barriers and facilitators based on demographic characteristics.** The association between barriers to ADR reporting and demographics was evaluated by using Mann-Whitney U and Kruskal Wallis test. The results are presented in Table 4, see Fig 4.

The majority of the male participants aged between 41–50 years agreed regarding the lack of knowledge and confidence, as well as lack of time and interest as a barrier ($p < 0.05$). Most participants between 31–40 years of age group or having 16–20 years of job experience agreed that legal liability can be a barrier to reporting an ADR ($p < 0.05$). Many participants aged

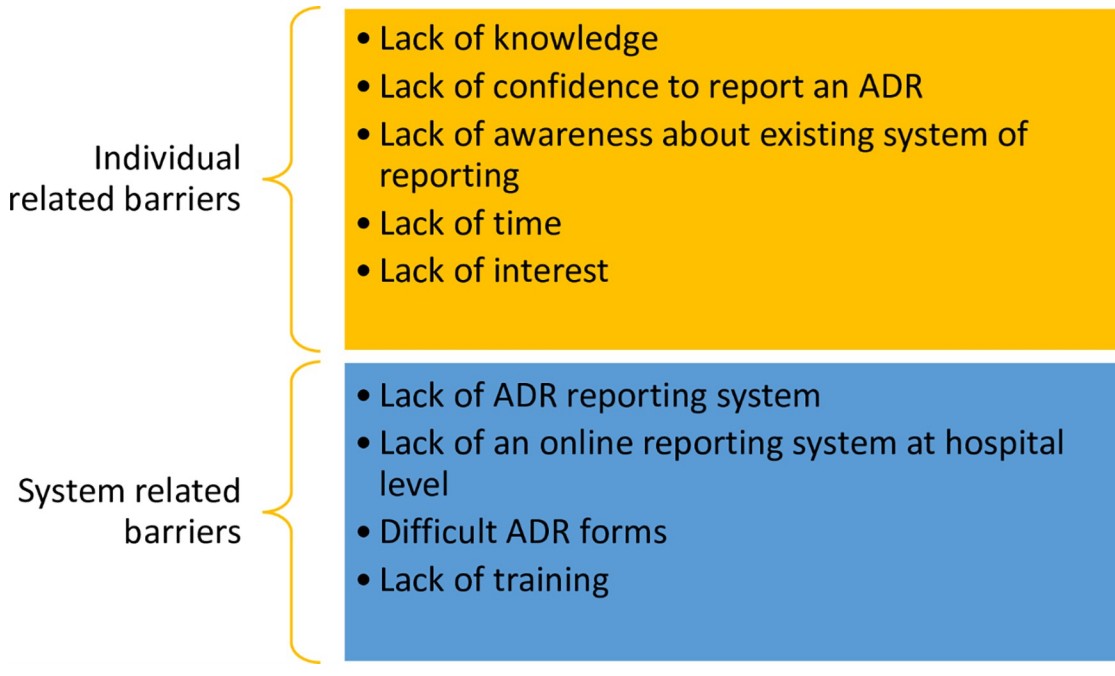

**Fig 1. Individual and system related barriers.**

between 31–35 years agreed that increased reminders from the National PV Centre (NPC), the establishment of an online system, continuous medical education, training, and educational seminars could improve ADR reporting (p<0.05) as given in Table 5 and presented in Fig 5.

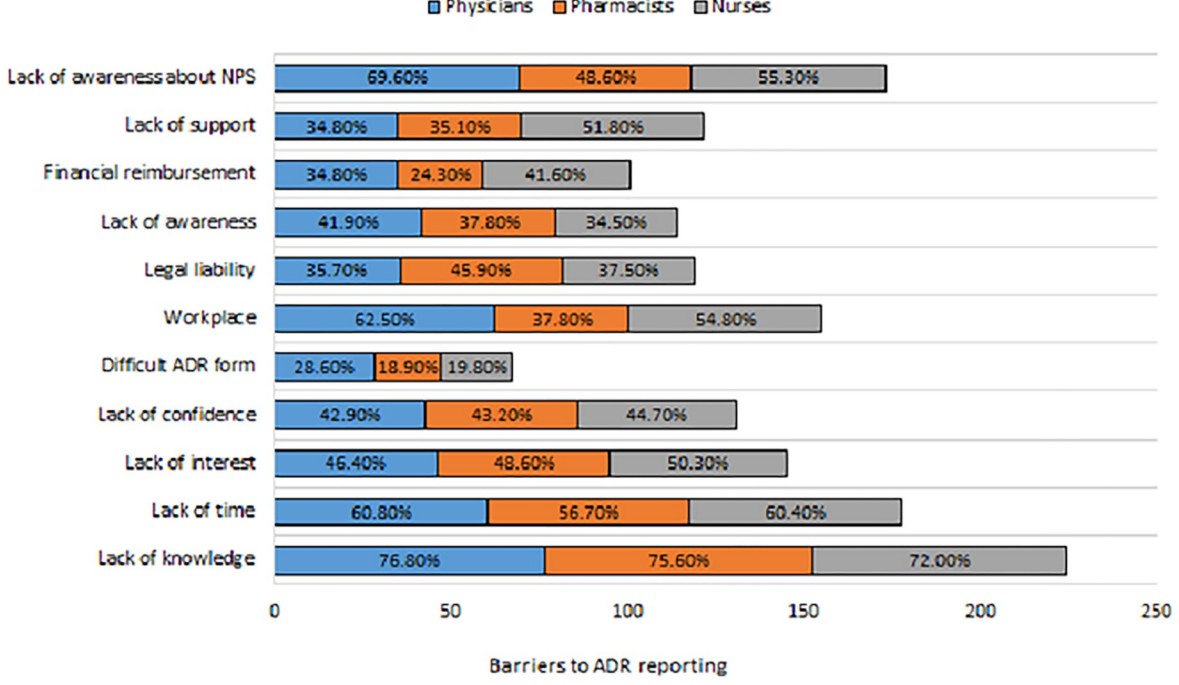

**Fig 2. Barriers to ADR reporting by HCPs.**

**Table 2. Barriers to ADR reporting by HCPs.**

| Barriers to ADR reporting | Category | Response n (%) | | |
|---|---|---|---|---|
| | | Disagree | Neutral | Agree |
| Lack of knowledge if an ADR happened | Physicians | 16(14.3) | 10 (8.9) | 86 (76.8) |
| | Pharmacists | 6(16.2) | 3 (8.1) | 28 (75.6) |
| | Nurses | 34(7.3) | 21 (10.7) | 142(72) |
| Lack of time for reporting | Physicians | 23(20.6) | 21 (18.8) | 68(60.8) |
| | Pharmacists | 11(29.7) | 5 (13.5) | 21 (56.7) |
| | Nurses | 44(22.4) | 34 (17.3) | 119(60.4) |
| Lack of interest to report about an ADR | Physicians | 35(31.2) | 25 (22.3) | 52(46.4) |
| | Pharmacists | 10(27) | 9 (24.3) | 18 (48.6) |
| | Nurses | 67(34) | 31 (15.7) | 99 (50.3) |
| Lack of confidence in discussing the ADRs with the prescriber | Physicians | 39(34.8) | 25 (22.3) | 48 (42.9) |
| | Pharmacists | 8(21.6) | 13 (35.1) | 16 (43.2) |
| | Nurses | 67(34) | 42 (21.3) | 88 (44.7) |
| The ADR form is too difficult to fill | Physicians | 48(42.8) | 32 (28.6) | 32 (28.6) |
| | Pharmacists | 26(70.3) | 4 (10.8) | 7(18.9) |
| | Nurses | 101(51.2) | 57 (28.9) | 39 (19.8) |
| Did not know how to report an ADR in my work place | Physicians | 13(11.6) | 29 (25.9) | 70 (62.5) |
| | Pharmacists | 16(43.2) | 7 (18.9) | 14(37.8) |
| | Nurses | 62(31.5) | 27 (13.7) | 108(54.8) |
| Fear of legal liability | Physicians | 34(30.3) | 38 (33.9) | 40(35.7) |
| | Pharmacists | 12(32.4) | 8 (21.6) | 17(45.9) |
| | Nurses | 67 (33.9) | 56 (28.4) | 74 (37.5) |
| Unaware of the need to report an ADR | Physicians | 32 (28.6) | 33 (29.5) | 47 (41.9) |
| | Pharmacists | 16 (43.2) | 7 (18.9) | 14 (37.8) |
| | Nurses | 79 (40.1) | 50 (25.4) | 68 (34.5) |
| Lack of financial reimbursement | Physicians | 43 (38.4) | 30 (26.8) | 39 (34.8) |
| | Pharmacists | 15 (40.5) | 13 (35.1) | 9(24.3) |
| | Nurses | 70 (35.6) | 45 (22.8) | 82 (41.6) |
| Lack of support from colleagues and administration | Physicians | 25(22.3) | 48 (42.9) | 39 (34.8) |
| | Pharmacists | 10 (27) | 14 (37.8) | 20(35.1) |
| | Nurses | 59 (30) | 36 (18.3) | 102(51.8) |
| Unaware of the existence of a national ADR reporting system | Physicians | 18 (16.1) | 16 (14.3) | 78(69.6) |
| | Pharmacists | 11 (29.7) | 8 (21.6) | 18 (48.6) |
| | Nurses | 50 (25.4) | 38 (19.3) | 109(55.3) |

## Discussion

The current study has identified barriers and facilitators among HCPs regarding ADR reporting in Pakistan. The respondents' experiences during the process of ADR reporting have also been highlighted.

The results of the study indicated a lack of knowledge if an ADR happened, a lack of confidence in discussing an ADR report among HCPs, and a lack of awareness about local settings and policies of PV. These were considered significant barriers experienced by many HCPs. Similar barriers were also identified by other studies, whereby the majority of pharmacists were unaware of the unavailability of reporting forms, as well as reporting procedures, or they did not report [32, 33]. Therefore, by providing education and training to HCPs, active participation in PV activities can be encouraged. This was evident from the studies conducted

**Table 3. Facilitators to ADR reporting by HCPs.**

| Facilitators to ADR reporting | Category | Response n (%) | | |
|---|---|---|---|---|
| | | **Disagree** | **Neutral** | **Agree** |
| Extra time should be given to report ADRs (other than duty hours) | Physicians | 0 (18) | 0 (18) | 112(100.0) |
| | Pharmacists | 0 (18) | 0 (18) | 37 (100) |
| | Nurses | 114 (57.9) | 18 (9.1) | 65 (33) |
| Incentives | Physicians | 0 (18) | 0 (18) | 112(100.0) |
| | Pharmacists | 0 (18) | 0 (18) | 37 (100) |
| | Nurses | 121 (61.4) | 29 (14.7) | 47 (23.9) |
| Continuous medical education, training related to ADR reporting | Physicians | 32 (28.6) | 0 (18) | 96 (71.4) |
| | Pharmacists | 0 (18) | 0 (18) | 37 (100.0) |
| | Nurses | 0 (18) | 0 (18) | 197 (100) |
| Reminders and increased awareness from the ADR Monitoring Centre | Physicians | 16 (14.3) | 3 (2.7) | 93 (83.1) |
| | Pharmacists | 0 (18) | 0 (18) | 37 (100) |
| | Nurses | 88 (44.7) | 4 (2.0) | 105 (53.3) |
| Online system for ADR reporting should be available | Physicians | 0 (18) | 10 (8.9) | 102 (91.1) |
| | Pharmacists | 6 (16.2) | 7 (18.9) | 24 (64.9) |
| | Nurses | 10 (5) | 5 (2.5) | 182 (92.4) |

among HCPs from Egypt, Nigeria, India, Nepal, and Pakistan, where an improvement in the knowledge, attitude, and practices were observed following an educational intervention promoting ADR reporting [34–38].

Lack of time to report an ADR was highlighted as an important barrier by the majority of the HCPs. This could be due to overly occupied staff in hospitals and workforce shortages. There were 195,896 physicians, 99228 nurses, and 32511 pharmacists in Pakistan's public

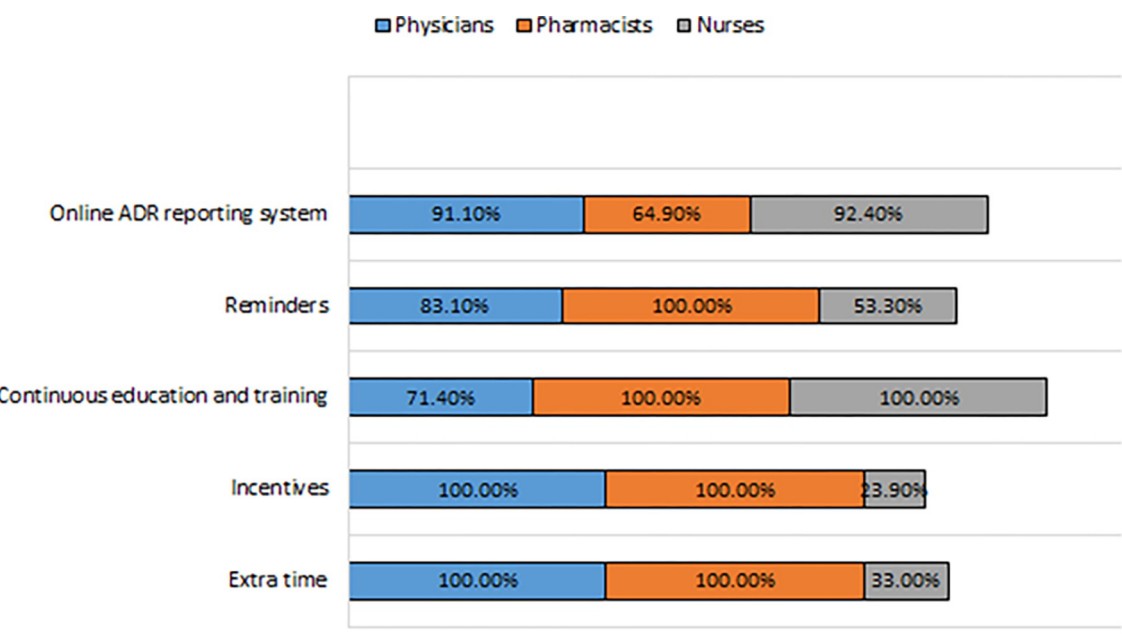

**Fig 3. Facilitators to ADR reporting by HCPs.**

**Table 4. Significance of ADRs related barriers based on demographic characteristics.**

| Barriers to ADR reporting | p-value | | | | |
|---|---|---|---|---|---|
| | Category | Gender | Age | Qualification | Experience |
| Lack of knowledge if an ADR happened | 0.249 | *0.017* | *0.000* | 0.689 | *0.009* |
| Lack of time for reporting | 0.890 | *0.006* | *0.011* | 0.838 | *0.027* |
| Lack of interest to report about an ADR | 0.816 | 0.958 | *0.028* | 0.894 | 0.125 |
| Lack of confidence in discussing the ADRs with the prescriber | 0.815 | 0.480 | *0.016* | 0.810 | 0.139 |
| The ADR form is too difficult to fill | *0.002* | 0.081 | 0.491 | 0.896 | 0.150 |
| Unaware of the existence of a national ADR reporting system | *0.009* | 0.269 | 0.604 | 0.374 | 0.908 |
| Did not know how to report an ADR in my workplace | *0.001* | 0.670 | *0.000* | 0.508 | 0.066 |
| Fear of legal liability | 0.610 | 0.258 | *0.005* | 0.247 | *0.035* |
| Unaware of the need to report an ADR | 0.158 | 0.724 | 0.153 | 0.456 | 0.347 |
| Lack of financial reimbursement | 0.312 | *0.027* | 0.070 | 0.720 | 0.761 |
| Lack of support from colleagues and administration | 0.426 | 0.354 | *0.014* | 0.693 | 0.205 |
| Difficulties to report an ADR when patients are treated with several drugs | 0.176 | 0.289 | 0.083 | 0.665 | 0.596 |
| I never get back any feedback on what action is taken from PV centre | 0.945 | 0.490 | 0.200 | 0.968 | 0.302 |

sector hospitals [10]. According to an estimate, the population to health services ratio for physicians is 0.82 (while the standard physician to population ratio is 1:1000), for nurses and midwives 0.57 per 1000 population (the standard ratio is 3 nurses per 1000), and 0.9 for pharmacists per 100,000 population (whereas the WHO recommended ratio is 1 pharmacist per 2000 population) [39, 40].

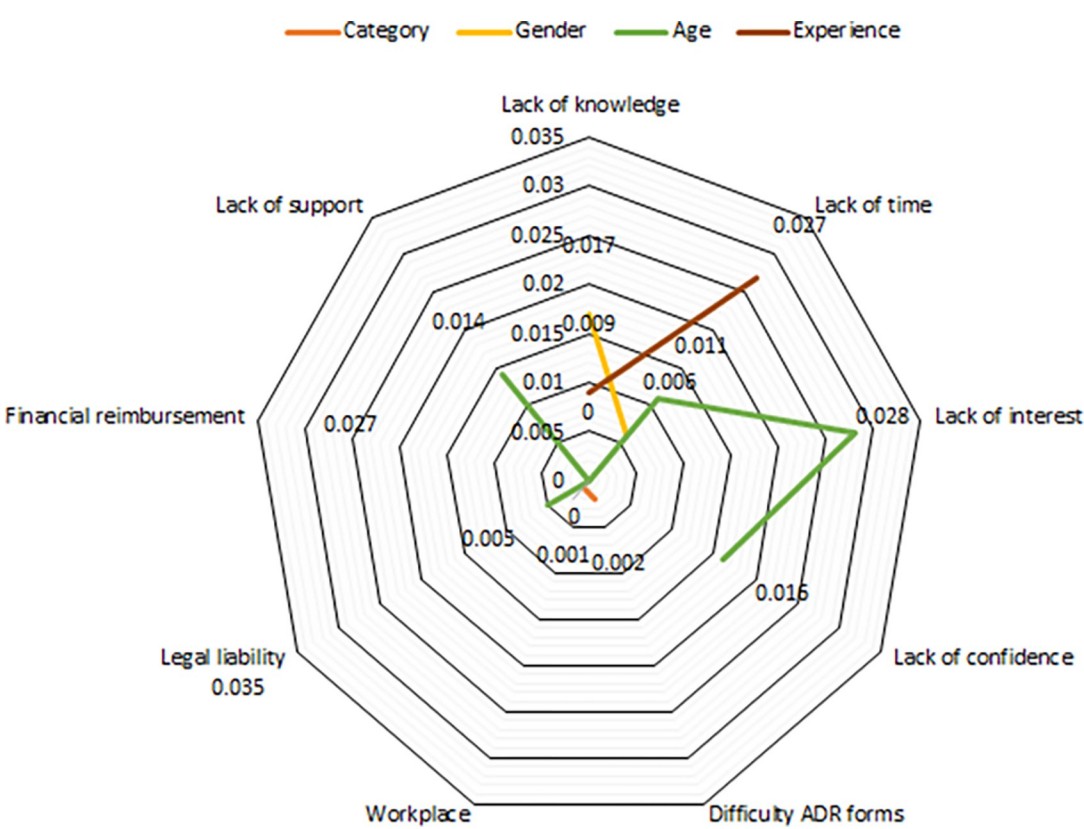

**Fig 4. Analysis of barriers based on demographic characteristics.**

**Table 5. Significance of ADRs related facilitators based on demographic characteristics.**

| Facilitators to ADR reporting | p-value | | | | |
|---|---|---|---|---|---|
| | Category | Gender | Age | Qualification | Experience |
| Extra time should be given to report ADRs (other than duty hours) | *0.000* | *0.000* | 0.205 | 0.685 | 0.956 |
| Incentives | *0.000* | *0.000* | 0.702 | 0.317 | 0.824 |
| Continuous medical education, training related to ADR reporting | *0.000* | 0.704 | *0.000* | 0.498 | 0.135 |
| Reminders and increased awareness from the ADR Monitoring Centre | *0.000* | 0.851 | *0.000* | *0.000* | *0.000* |
| Online system for ADR reporting should be available in all public hospitals | 0.131 | 0.053 | 0.494 | *0.002* | 0.576 |

This low ratio of HCPs to patients could result in an increased workload for the rest of HCPs, ultimately leading to the lack of time to report an ADR. The workforce shortages are also illustrated by the fact that every year many HCPs including physicians, pharmacists, and nurses move overseas to look for better opportunities. According to an estimate, around 17.6% of physicians and 15% of nurses migrated to developed countries for better working opportunities, which contributes towards the shortage of skilled healthcare workers leading to the extra workload on the existing HCPs [41]. Improvement can be made if the government pays proper attention to the sufficient staffing of HCPs in the hospitals. This could not only decrease the time for the reporting of a suspected ADR but also could help reduce the chance of making errors during the medication use process [42].

Lack of confidence in discussing a suspected ADR with other colleagues is another major barrier. The studies have identified that a lack of confidence in determining whether a drug has caused an ADR or otherwise could also be a determinant of underreporting [27, 43–45]. A study by Gupta et al. (2018) showed that 7.5% of participants including pharmacists and nurses identified a lack of confidence to report an ADR as a barrier [46]. Similarly, Terblanche et al. (2017) found that 22% of the HCPs were not sure to speak about whether an ADR has

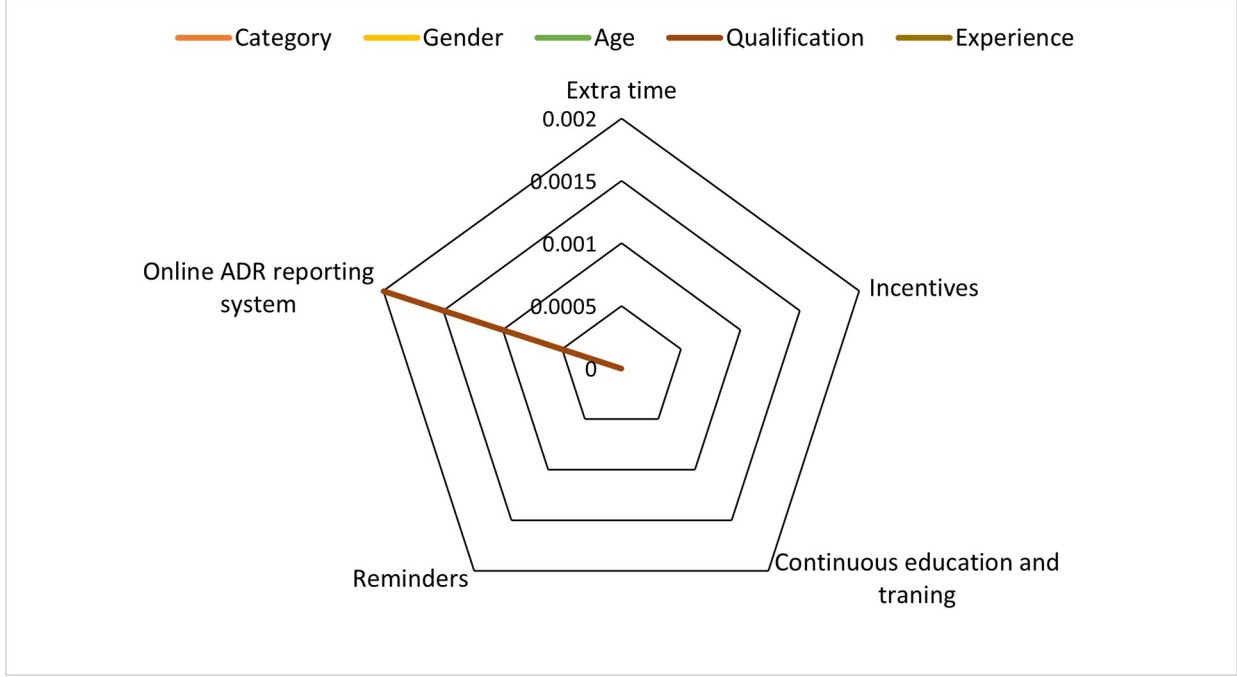

**Fig 5. Analysis of facilitators based on demographic characteristics.**

happened or not, hence they never reported an ADR [47]. This could be due to the weakness of the health system, as HCPs seem to have less trust and confidence in sharing ADR data and information.

Regarding health system-related barriers, the majority of HCPs identified a lack of reporting systems as a barrier to the reporting of an ADR. Being unable to have access to the online system may deter the HCPs to establish the possibility that an ADR has happened or otherwise [48, 49]. Moreover, the tertiary care public hospitals in Pakistan are in the process of shifting from manual to automated systems, and therefore both awareness and access to online ADR reporting are still not prevalent among HCPs [29]. The DRAP has recently launched a web portal for ADR reporting and a mobile application known as Med Safety application for the reporting of ADRs in collaboration with the United Kingdom (20) Medicines and Healthcare products Regulatory Authority [50], Uppsala Monitoring Centre (UMC) and the WHO normally collects and disseminate data about the safety-related issues of the drugs [51]. However, it is too early to establish if ADR reporting has improved after the launch of this mobile application or otherwise.

In many developed countries including France, Sweden, the Netherlands, and the UK, the ADR reporting rate was found to be 40–70% among physicians [50, 52]. But, in countries like Pakistan, a lack of awareness about reporting systems barred the reporting of suspected ADRs. Thus, one of the reasons for not reporting an ADR is the lack of awareness about local or national PV system [53].

Most participants in our survey did not emphasize lack of financial reimbursement or fear of legal liability as a barrier to ADR reporting. However, this has been indicated by other studies. Rather, it was noted from the literature that non-monetary incentives could be a possible option for encouraging the HCPs about ADR reporting in their work setting [54–56].

In the UK, the undergraduate courses cover the Yellow Card scheme in all medical schools and the students are being taught about the Yellow Card scheme [57]. This leads to having a better reporting system with a greater number of ADR report cases. To resolve the problem of underreporting, there is a need to update and improve the undergraduate syllabus of these key healthcare professions by making reporting of ADRs and causality assessment a mandatory part of the syllabus [53]. Such an issue can be resolved by involving the relevant councils which are authorized in the update of the curriculum and for the necessary certification.

Many countries have an automated patient data management system, which could raise a red flag during the dispensing of a prescription with a possible chance of an ADR. In Pakistan, the hospital management system and patient data records are facing a shift from being manual to being automated [58]. Hence, along with the advancements in the hospital management system, the participants also strongly recommended the establishment of an online system in each public hospital. This online system could be readily accessible to all HCPs. Continuous medical or pharmacy education, training, and seminars were identified as key facilitators to overcoming the problem of underreporting. Thus, to improve the reporting practices, the HCPs also suggested that they would like to receive additional training regarding medicines safety and ADR reporting in the form of seminars or workshops as well as newsletters and reminders from national PV centres in Pakistan. The WHO drug safety unit is providing many online courses and training sessions free of cost. These could serve as learning resources for HCPs who are unable to join such activities in person. Besides, the provincial drug control unit of Punjab is monthly publishing a newsletter on the safety of medicines to inform the recent developments in drug safety [59].

### Strengths and limitations of the study

The study has some limitations, such as, that we used self-reported responses as the method of inquiry, which could have led to the recall and social desirability bias. However, as the self-administered questionnaire did not include participants' names and data were anonymously recorded, thus the potential for social desirability was reduced.

The cross-sectional study design selection may have led to gaps, which could be otherwise filled if used a time series analysis to explain the factors responsible for ADR underreporting. Additionally, there was an over-representation of nurses and physicians over pharmacists, owing to the current healthcare team dynamics in the Pakistani healthcare setting, whereby pharmacists are still under-recruited in public hospitals.

The findings of the study cannot be generalized to the whole country as this study was conducted among HCPs from Lahore, Punjab. Nevertheless, the study was conducted in the second-largest city, Lahore with more advanced and facility-based systems. Thus, despite these limitations, the information is still useful for both HCPs as well as for policymakers.

A key strength of this study was the inclusion of physicians, pharmacists, and nurses working in tertiary care public hospitals. This helped to identify the range of challenges encountered in most facility-based healthcare settings. The study has also generated key insights on the barriers and facilitators to improving the overall PV system in the country.

## Conclusion

Healthcare professionals including physicians, pharmacists, and nurses have identified a perceived lack of knowledge, time, confidence, and support from colleagues about the national PV system as the barriers to ADR reporting. Extra time for ADR reporting, incentives, training and education, reminders, and availability of an online ADR reporting system were identified as possible facilitators for improvement.

## Supporting information

**S1 Questionnaire.**
(DOCX)

**S2 Questionnaire. Inclusivity in global research.**
(DOCX)

## Acknowledgments

The authors are extremely grateful to all HCPs for their cooperation in completing this survey.

## Author Contributions

**Conceptualization:** Rabia Hussain, Mohamed Azmi Hassali.

**Data curation:** Rabia Hussain, Tayyaba Akram.

**Formal analysis:** Rabia Hussain.

**Investigation:** Rabia Hussain, Anees ur Rehman.

**Methodology:** Rabia Hussain, Mohamed Azmi Hassali, Zaheer-Ud-Din Babar.

**Project administration:** Rabia Hussain.

**Resources:** Rabia Hussain, Furqan Hashmi, Zaheer-Ud-Din Babar.

**Software:** Rabia Hussain, Tayyaba Akram.

**Supervision:** Mohamed Azmi Hassali, Zaheer-Ud-Din Babar.

**Validation:** Jaya Muneswarao.

**Writing – original draft:** Rabia Hussain, Tayyaba Akram, Anees ur Rehman.

**Writing – review & editing:** Jaya Muneswarao, Zaheer-Ud-Din Babar.

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
