## [Decision Letter · Decision Letter 0]

30 Dec 2021

PONE-D-21-35762Barriers and facilitators to adverse drug reaction reporting by healthcare professionals: a cross-sectional surveyPLOS ONE

Dear Dr. Hussain,

Thank you for submitting your manuscript to PLOS ONE. After careful consideration, we feel that it has merit but does not fully meet PLOS ONE’s publication criteria as it currently stands. Therefore, we invite you to submit a revised version of the manuscript that addresses the points raised during the review process.

We look forward to receiving your revised manuscript.

Kind regards,

Vijayaprakash Suppiah, PhD

Academic Editor

PLOS ONE

Reviewers' comments:

Reviewer's Responses to Questions

**Comments to the Author**

1. Is the manuscript technically sound, and do the data support the conclusions?

Reviewer #1: Partly

Reviewer #2: Partly

2. Has the statistical analysis been performed appropriately and rigorously? 

Reviewer #1: Yes

Reviewer #2: Yes

3. Have the authors made all data underlying the findings in their manuscript fully available?

Reviewer #1: Yes

Reviewer #2: Yes

4. Is the manuscript presented in an intelligible fashion and written in standard English?

Reviewer #1: No

Reviewer #2: Yes

5. Review Comments to the Author

Reviewer #1: Thank you for the opportunity to review the manuscript entitled “Barriers and facilitators to adverse drug reaction reporting by healthcare professionals: a cross-sectional survey”. The authors have conducted a questionnaire-based study to determine the barriers and facilitators related to adverse drug reaction (ADR) reporting of physicians, pharmacists and nurses in Lahore, Pakistan. Although assessing the knowledge, attitude and practice of the healthcare professionals (HCPs) in ADR reporting would have been more appropriate for a country with infancy stage of pharmacovigilance activity, the authors have identified the barriers and facilitators of the ADR reporting. Please refer to attachment for detailed comments.

Reviewer #2: This is a study looking into the almost universal issue of ADR underreporting but from the context of health professionals (physicians, pharmacists, nurses) working in hospitals in Lahore.

In your introduction (line 117-118) you say there is a scarcity of literature from Pakistan but there are a number of papers on this topic (e.g. https://doi.org/10.1177/0018578720910401) although most focus only on physicians and / or pharmacists. The introduction should be more open about what work is already out there on this topic and therefore what your study adds.

Abstract:

I feel this would benefit from a review. The way it is written is a bit confusing in places (e.g. line 43-45 "similar to that" but then the information presented does not relate to the prior sentence at all). Also the double negatives in line 42-43 are confusing. Perhaps add some basic information re distribution by HCP role. Rather than presenting % for some HCP for some barriers / facilitators focus more on headline messages, including any differences on basis of demographics. Make sure you include reference to limitations of the study to help the reader understand how to interpret the findings.

Introduction:

Line 89-90 Better to reframe this around spontaneous reporting as this is what the rest of the section seems to be focussed on rather than "spontaneous reporting, post marking surveillance etc" as there is some overlap which affects clarity of message.

Please ensure you have referenced all statements as required e.g. ref for line 82-83 re 5th cause of death, Pirmohamed's paper from line 89-90 (same is true in discussion, see below).

It would be helpful for international readers to have more context re ADR reporting in Pakistan e.g. which health professionals can report, can patients report, how are reports submitted, etc. Please add a bit more information regarding this.

Aims/objectives could be presented in a more explicit way (very minor amendment needed)

In the methods there needed to be more information regarding sampling / recruitment and data collection. How were potential participants identified and approached for recruitment (just says convenience sampled but no further information) - was there any attempt at stratification by HCP role for example? How was the survey administered - was it online / paper / self-administered etc. Were any reminders sent or was it a one-off of who was available on the day? Also was there any space for free-text comments or were all questions based on Likert scale / tick-boxes?

In results, line 158 no need to say 384 twice

Throughout the results there is a tendency to present the same data in detail in the text and in a table and in figures - this is overkill - a summary of key headline messages in the text then either a table OR figure is adequate and reduces confusion.

I'm not sure the IQR / median add anything given the use of a 3 point scale. Tables would be clearer without them.

Figure 4 (line 490) is labelled as Figure 2

Figure 3 and 4 take different styles despite presenting equivalent data. If you plan to retain figures please consider consistency of style.

Tables 3 and 4 I found quite hard to pull out the key data. It would be clearer to present just the significant differences and add a statement to say all others were non-sig. Explain 'category' means type of HCP. consider if you can include the direction of difference within the table rather than having to jump back and forth from table to text to see where the difference lay.

Please review the discussion to ensure all studies mentioned are referenced e.g. lines 280-282, 286

250-51 you say that providing education would increase active participation - studies have shown this link so please provide evidence in the form of some reference(s).

Time is a major barrier reporting in most studies of this type - I found the section 252-265 to be confusing as it took a long time to work out that this was providing an explanation for the barrier of time - make it more explicit upfront but also raise the fact that this comes up almost universally in studies re barriers to reporting (i.e, this is not unique or only related to the issue of workforce moving abroad).

266-272 you discuss confidence yet confidence in your study seemed lower than in other papers - why might this be?

276-279 interesting re the app - is there any evidence yet that it has improved reporting? If not then just add that it's too early to see the results.

280-282 ADR reporting is NOT mandatory in the UK for physicians, although strongly encouraged (and I don't believe this is the case in some of the other countries) - where did you get this information from?

Please add some critical commentary around limitations. The conclusion should be a bit more focussed to make clear the context and any limitations which should be borne in mind while considering these conclusions.

6. PLOS authors have the option to publish the peer review history of their article (what does this mean?). If published, this will include your full peer review and any attached files.

Reviewer #1: No

Reviewer #2: No

---

## [Author Response · Author response to Decision Letter 0]

22 Feb 2022

Reviewer 1

Thank you for the opportunity to review the manuscript entitled “Barriers and facilitators to adverse drug reaction reporting by healthcare professionals: a cross-sectional survey”. The authors have conducted a questionnaire-based study to determine the barriers and facilitators related to adverse drug reaction (ADR) reporting of physicians, pharmacists and nurses in Lahore, Pakistan. Although assessing the knowledge, attitude and practice of the healthcare professionals (HCPs) in ADR reporting would have been more appropriate for a country with infancy stage of pharmacovigilance activity, the authors have identified the barriers and facilitators of the ADR reporting.

Overall, the manuscript contains many grammatical errors and would greatly benefit from proofreading and language editing. 

Abstract

1. Page 2, Line 33: Abbreviate adverse drug reaction at first mention.

Response: Thank you. It’s corrected now. Line 33

2. There is no mention of facilitators of ADR in the background and objective.

Response: We have amended the text according to the suggestions. Line 115-117 & 123-125

3. The objective mentioned in the abstract and in the main text is not the same. The objective in the abstract includes identifying possible solutions to improve the underreporting of ADRs among healthcare professionals from Lahore, Pakistan. However, the objective in the main text does not outline this component. 

Response: Thank you, we have corrected the text as suggested. Line 34-36 & 123-125

4. Page 2, Line 54: Lack of knowledge or awareness (as mentioned in line 47)? Standardized use of keywords should be followed.

Response: Corrected, we have changed the word to lack of knowledge and made it uniform throughout the text. Line 46

Introduction

1. Page 4, Line 80: The word adverse drug reaction should have been abbreviated at first mention in Line 80 if abbreviation in abstract not considered as the first mention. The abbreviation and full name have been used interchangeably throughout the manuscript. Standardize the use of abbreviation (where appropriate) after first mention.

Response: It was correctly abbreviated at line 58.

2. Page 4, Line 86 & 87: Include reference(s).

Response: Thank you. We have amended the information and references are added. Line 66-68

3. Page 5, line 101-102: Grammar correction “To date, there are a total of 171 member countries for the PIDM…”

Response: Many thanks, it has been corrected now. Line 81

4. Page 5, Line 104: PIDM already been abbreviated at first mention (line 99). Delete the full name and use abbreviation instead.

Response: Many thanks. It was corrected in line 83 and made uniform throughout the text.

5. Include ADR prevalence data in Pakistan.

Response: Thank you, we have added the information in Line 100-103, “However, since the establishment of DRAP, a total of 6587 ADR reports have been received by the National Pharmacovigilance Centre.”

6. Include data on the knowledge, attitude and practice towards ADR reporting of HCPs in Pakistan.

Response: Thank you. the information is added to Line 110-115

“Studies from Pakistan have summarized that HCPs had poor knowledge about PV and ADR reporting and identified lack of awareness about PV guidelines and absence of a PV center as barriers to ADR reporting. As Iffat et al. (2014), Nisa et al. (2018) and Syed et al. (2018) indicated that HCPs had poor knowledge about ADR reporting systems and the unavailability of incentives and professional environment hindered the ADR reporting (20-22).”

7. Include evidence to establish that under-reporting of ADR is an issue in Pakistan.

Response: Thank you, we have added the information to Line 90-106

“The medicines safety issue has a wider impact on the country as big as Pakistan with a population of over 200 million people (15). The incident at Punjab Institute of Cardiology (PIC) in Lahore, which accounted for the life of over a hundred patients, highlighted the drug safety monitoring issues in the country including negligence on the part of healthcare professionals (16). Post the PIC incident, the Drug Regulatory Authority of Pakistan has developed Pakistan’s national pharmacovigilance system. Under the guidelines of DRAP, each healthcare professional, patient or even a caregiver can report about an ADR, which is being provided through an online submission system known as MED Vigilance E-Reporting System. This is meant to support ADR reporting both from the healthcare professionals, as well as from the patients (17). These reports are then analysed and further communicated to the Uppsala PIDM for signal detection (13). However, since the establishment of DRAP, a total of 6587 ADR reports have been received by the National Pharmacovigilance Centre and only 124 reports had been made by the pharmacovigilance center in Punjab. However, none of the reports was made by the public (18). To date, the ADR reporting rate in Pakistan is suboptimal and does not qualify for the WHO standard for ADR reporting, which is 200 ADR reports per million inhabitants in a year (19). Therefore, the underreporting of ADR remains a major concern in almost all parts of the country.” 

Method

1. Page 6, Line 129: Healthcare or health care? Use standardized term.

Response: Thank you, We have standardized the term as healthcare.

2. The eligibility criteria of the respondents could be described in further details. Were trainees included? How about those who were on long leave during the data collection period?

Response: A detailed description of methods has been added to the text. Line 131-138 

“Study Population & Study Site

All healthcare professionals (including pharmacists, doctors, and nurses) working in the government tertiary care public hospitals were considered eligible to participate in the study. This was considered if they were registered with the relevant provincial/national council and willing to provide written consent to the researcher. Those who refused or were not willing to participate in the study were excluded. The selected tertiary care public hospitals were the representative hospitals of Punjab and almost served patients from all areas of the province and were well equipped with modern facilities.

3. Include a brief description of the study site(s).

Line 136-138

The selected tertiary care public hospitals were the representative hospitals of Punjab and almost served patients from all areas of the province and were well equipped with modern facilities.”

4. Page 7, line 145: HCP was not abbreviated at first mention.

Response: Thank you. We have removed the abbreviation and used healthcare professionals throughout the text.

5. Include information on the reliability and validity of the questionnaire.

Response: Many thanks. Information is added to Line 149-155

6. Was a pilot testing of the questionnaire conducted? 

Response: Yes, it was conducted and details are added to the text. Line 149-155

“Validation of survey questionnaire

The questionnaire was tested for its face and content validity and two academics from the School of Pharmaceutical Sciences, Universiti Sains Malaysia (USM) reviewed the questionnaire in terms of its clarity and relevance and was modified according to the suggestions provided. Furthermore, prior to the actual survey implementation, the questionnaire was pilot tested on 30 HCPs. These HCPs were later excluded from the study. The internal consistency of the questionnaire was found to be 0.72.”

7. Include information on the distribution of the questionnaire. Was a hardcopy or softcopy version used? How much time was given for the respondents to return the completed questionnaire? What are the measures taken to increase response rate?

Response: Yes, the details are added to the methods section. Line 156-167

“A face-to-face survey was administered using questionnaires along with the explanatory statement about the research project and consent form in a hard copy.

Questionnaires were provided with a written informed consent form which clearly assured the participants about the confidentiality and anonymity of the gathered information. The self-administered questionnaires were distributed to HCPs after obtaining permission from the relevant head of the department, nursing heads, and chief pharmacists. The majority HCPs filled and returned the questionnaires on the same day while remaining questionnaires were collected during periodic visits to the participating hospitals/ departments and reminders were also sent after a week, if deemed necessary. Extra copies of questionnaires were provided in order to avoid any inconvenience due to shortage/loss of questionnaires and no incentives were given to fill the questionnaires.”

8. Page 7, Line 154: Under-reporting or underreporting? Use standardized term.

Response: Thank you. We have standardized the term throughout the text.

9. Page 7, Line 154: Delete repeated word “and and” 

Response: Thank you for pointing it out.

“and” is omitted. Line 181

Results

1. Page 7 & 8, Line 161 – 168: The information presented seem redundant as already summarized in Table 1.

Response: Thank you for pointing this out. The text was removed. Line 188

2. As the level of knowledge of the HCPs were not assessed, the reported lack of knowledge seems to be what was perceived by the HCPs as opposed to the actual scenario. The term “perceived lack of knowledge” maybe more suitable to be used in the results, discussion and conclusion section.

Response: Thank you. It was rephrased as perceived lack of knowledge. Line 340

3. Suggest to combine Figure 1A, 1B and 1C to present the overall results and comparison of the barriers related to ADR reporting by healthcare professional category. The same may be done to present the data on the facilitators to ADR reporting.

Response: Many thanks. The figure was adjusted according to the suggestion. Fig 2 & 3

4. Overall, the inferential analysis data may be summarized and presented to complement the data in the Fig.3 and 4. 

Response: Thank you, it was presented in Fig 4 & 5.

Discussion

1. Page 11, Line 242-244: This information should be in “Results” section.

Response: Many thanks. Information was added to the Results section in Line 189-193

2. Page 12, Line 267-268: Include more references to denote the plurality in the statement “Studies have identified that a lack of confidence in determining whether a drug has caused a reaction also adds to one of the determinants of under-reporting (24).”

Response: Many thanks for pointing this out. Relevant references have been added to the section in Line 266-267

3. Page 12, Line 273-274: What is meant by “identified a lack of reporting system as a barrier hindering the reporting of an ADR”? Lack of system or lack of awareness to system? 

Response: Many thanks for pointing it out. Here, we are meant to state that lack of reporting system, because Pakistani hospitals are in transit of advancements, however, the problem related to the lack of a system still persists, as this shift is not uniform among all the tertiary care hospitals. Line 275-280

4. Page 12, Line 274: The word “barrier” or “hindering” should be used. Not both.

Response: Thank you, we have removed the word “hindering” from the text. Line 276

5. Page 13, Line 289-290: The lack of knowledge, skill and training was not assessed in this study making the statement “The current study has identified the lack of knowledge, skills and training of study participants, especially among physicians and nurses” invalid.

Response: Many thanks, We have removed this part.

6. Page 14, Line 313: WHO was not abbreviated at first mention.

Response: Many thanks, We have abbreviated it at the first mention in Line 82-83

7. What are the strength and limitations of the study?

Response: Many thanks, We have added to the text in line 320-336.

“Strengths and limitations of the study

The study has some limitations, such as, we used self-reported responses as the method of inquiry, which could have led to the recall and social desirability bias. However, as the self-administered questionnaire did not include participants’ names and data were anonymously recorded, thus the potential of social desirability was reduced. 

The cross-sectional study design selection may have led to many gaps, which could be otherwise filled if used a time series analysis to explain the factors responsible for ADR underreporting. Additionally, there was an over-representation of nurses and physicians over pharmacists, owing to the current healthcare team dynamics in the Pakistani healthcare setting, whereby pharmacists are still under-recruited in public hospitals. 

The findings of the study cannot be generalized to the whole country as this study was conducted among HCPs from Lahore, Punjab. However, the study was conducted in the second-largest city, Lahore with more advanced and facility-based systems. Thus, despite these limitations, the information is still useful for both healthcare professionals as well as for policymakers.

A key strength of this study was the inclusion of physicians, pharmacists, and nurses working in tertiary care public hospitals. This helped to identify the range of challenges encountered in most facility-based healthcare settings. Being the study is important as it has generated key insights on the barriers and facilitators to improve the overall pharmacovigilance system in the country.”

Reviewer 2

Abstract:

Q. I feel this would benefit from a review. The way it is written is a bit confusing in places (e.g. line 43-45 "similar to that" but then the information presented does not relate to the prior sentence at all). Also the double negatives in line 42-43 are confusing. Perhaps add some basic information re distribution by HCP role. Rather than presenting % for some HCP for some barriers / facilitators focus more on headline messages, including any differences on basis of demographics. Make sure you include reference to limitations of the study to help the reader understand how to interpret the findings.

Response: Many thanks. The abstract is revised, and double negatives are removed. Limitations and strengths are also added to the manuscript.

Abstract

“The timely reporting of adverse drug reactions (ADRs) could improve pharmacovigilance in a healthcare system. However, barriers to the underreporting of ADRs exist in almost all healthcare systems. The objective of the study was to identify the barriers and facilitators regarding pharmacovigilance (PV) activities from healthcare professionals’ (HCPs) point of view in Lahore, Pakistan. A cross-sectional questionnaire-based survey was used between September 2018 to January 2019, and the data was collected through convenience sampling of physicians, pharmacists, and nurses at tertiary care public hospitals of Lahore. A total of 384 questionnaires were distributed, and 346 HCPs responded to the survey. Over sixty-two percent of physicians and 54.8% of nurses agreed that they did not know how to report an ADR in their workplace. About 43.2% of pharmacists and 40.1% of nurses disagreed that they were not aware of the need for ADR reporting. Similarly, 41.6% of nurses identified a lack of financial reimbursement and 51.8% highlighted lack of support from a colleague as a barrier for the underreporting of ADR. The majority of participants, including 69.6% physicians, 48.6% pharmacists, and 55.3% nurses identified the lack of knowledge about the existence of a national PV center. Extra time for ADR reporting, incentives, continuous medical education, reminders, and availability of an online ADR reporting system was classed as the facilitators and were agreed upon by the majority of HCPs.” 

Introduction:

Q. Line 89-90 Better to reframe this around spontaneous reporting as this is what the rest of the section seems to be focussed on rather than "spontaneous reporting, post marking surveillance etc" as there is some overlap which affects clarity of message.

Response: Thank you. It has been rephrased. Line 69-77

Please ensure you have referenced all statements as required e.g. ref for line 82-83 re 5th cause of death, Pirmohamed's paper from line 89-90 (same is true in discussion, see below).

Response: Many thanks, references have been added. Line 64-68

Q. It would be helpful for international readers to have more context re ADR reporting in Pakistan e.g. which health professionals can report, can patients report, how are reports submitted, etc. Please add a bit more information regarding this.

Response: Many thanks, we have added the relevant information to the text. Line 92-98

“Post the PIC incident, the Drug Regulatory Authority of Pakistan has developed Pakistan’s national PV system. Under the guidelines of DRAP, each healthcare professional, patient or even a caregiver can report about an ADR, which is being provided through an online submission system known as MED Vigilance E-Reporting System. This is meant to support ADR reporting both from the HCPs, as well as from the patients (17). These reports are then analyzed and further communicated to the Uppsala PIDM for signal detection (13).”

Q. Aims/objectives could be presented in a more explicit way (very minor amendment needed)

Response: Thank you, we have added the following text to the manuscript. Line 120-122

Response: This is the first study being conducted to determine the barriers and facilitators related to ADR reporting from all three cadres of HCPs including physicians, pharmacists, and nurses in Lahore, Pakistan.

Q. In the methods there needed to be more information regarding sampling / recruitment and data collection. How were potential participants identified and approached for recruitment (just says convenience sampled but no further information) - was there any attempt at stratification by HCP role for example? How was the survey administered - was it online / paper / self-administered etc. Were any reminders sent or was it a one-off of who was available on the day? Also was there any space for free-text comments or were all questions based on Likert scale / tick-boxes?

Response: Thank you, we have revised the Methods section. Line 123-167.

“Methods

Study Design and Study Period

A cross-sectional study was conducted to assess the barriers and facilitators among HCPs regarding PV activities in Lahore. The study was conducted from September 2018 to January 2019.

Study Population & Study Site

All HCPs (including pharmacists, doctors, and nurses) working in the government tertiary care public hospitals were considered eligible to participate in the study. This was considered if they were registered with the relevant provincial/national council and willing to provide written consent to the researcher. Those who refused or were not willing to participate in the study were excluded. The selected tertiary care public hospitals were the representative hospitals of Punjab and almost served patients from all areas of the province and were well equipped with modern facilities.

Sample Size Determination and Sampling

The sampling technique was convenience sampling including physicians, pharmacists, and nurses, whereby participants were selected according to their accessibility, convenience, and proximity.

The sample size was calculated to be 384 by employing the Cochrane formula (27).

Development of survey questionnaire

The questionnaire was developed based on the literature and exploratory interviews-based findings from physicians, pharmacists, and nurses during the first phase of the study (9, 16, 28, 29). The questions included demographic details, experiences of HCPs (HCPs) regarding barriers and facilitators towards ADR reporting in their work setting. 

Section one included demographic characteristics including gender, age, occupation, education etc. Section two covered barriers related to the ADR faced by HCPs while reporting an ADR. Section three covered areas related to facilitators improving ADR reporting-related activities. 

Validation of survey questionnaire

The questionnaire was tested for its face and content validity and two academics from the School of Pharmaceutical Sciences, Universiti Sains Malaysia (USM) reviewed the questionnaire in terms of its clarity and relevance and was modified according to the suggestions provided. Furthermore, prior to the actual survey implementation, the questionnaire was pilot tested on 30 HCPs. These HCPs were later excluded from the study. The internal consistency of the questionnaire was found to be 0.72

Data collection

A face-to-face survey was administered using questionnaires along with the explanatory statement about the research project and consent form in a hard copy.

Questionnaires were provided with a written informed consent form which clearly assured the participants about the confidentiality and anonymity of the gathered information. The self-administered questionnaires were distributed to HCPs after obtaining permission from the relevant head of the department, nursing heads, and chief pharmacists. The majority HCPs filled and returned the questionnaires on the same day while remaining questionnaires were collected during periodic visits to the participating hospitals/ departments and reminders were also sent after a week, if deemed necessary. Extra copies of questionnaires were provided in order to avoid any inconvenience due to shortage/loss of questionnaires and no incentives were given to fill the questionnaires.”

Q. In results, line 158 no need to say 384 twice

Response: Many thanks for pointing this out. We have removed it. Line 185

Q. Throughout the results there is a tendency to present the same data in detail in the text and in a table and in figures - this is overkill - a summary of key headline messages in the text then either a table OR figure is adequate and reduces confusion.

Response: Thank you, we have edited the results as suggested. Line 183-233

“Results

Demographics data

Based on sample size calculation, a total of 384 questionnaires were distributed to HCPs (considering the inclusion criteria). The returned questionnaires were collected by the researcher with a response rate of 346 (90.10%) for all participating HCPs. The detailed demographics of the participants are described in Table 1.

Barriers and facilitators related to ADR reporting

Broadly, the barriers to ADR reporting can be classified into two categories, comprising of healthcare system-related barriers and individual-related barriers as given in Fig 1 (30). 

Insert Fig 1. here

The barriers related to ADR reporting were evaluated on a three-point Likert scale and responses were presented as Fig 2 for physicians, pharmacists, and nurses respectively. The majority of the respondents strongly agreed to the different aspects as the barriers towards ADRs, such as 86 (76.8%) physicians, 28 (75.6%) pharmacists and 142 (72.0%) nurses identified a lack of knowledge about ADR. Similarly, 68 (60.8%) physicians, 21 (56.7%) pharmacists, and 119 (60.4%) nurses agreed on lack of time as a hindrance to ADR reporting. Among all, the majority of nurses identified a lack of interest to report an ADR and a lack of confidence in discussing an ADR with other colleagues as the reasons for underreporting of ADR. This has also been shown in Table 2.

Insert Table 2 here

Most participants disagreed that the ADR form is too difficult to fill, however, 70 (62.5%) physicians and nurses 108 (54.8%) agreed, that they did not know, how to report an ADR in their workplace and17 (45.9%) pharmacists and 74 (37.5%) nurses considered fear of legal liability as a barrier. About one-third of the physicians 47 (41.9%) agreed that they were unaware of the need to report an ADR. Also ,the majority of nurses have identified lack of financial reimbursement 82 (41.6%) and lack of support from colleagues and administration 102 (51.8%) as possible barriers to ADR reporting. The participants including 78 (69.6%) physicians, 18 (48.6%) pharmacists, and 109 (55.3%) nurses identified the lack of awareness about the existence of the national PV center as a barrier to the ADR reporting.

Regarding facilitators to ADR reporting, the majority of the nurses disagreed with certain facilitating factors for ADR reporting such as extra time and incentives but agreed that continuous medical education, reminders, and an online system for reporting as agreed by a majority of the physicians and pharmacists as shown in Fig 3.

Insert Table 3 here

Insert Fig 3 here

Analysis of ADR reporting related barriers and facilitators based on demographic characteristics

The association between barriers towards ADR reporting and demographics was evaluated by using Mann-Whitney U and Kruskal Wallis test. The results are presented in Fig 4. 

Insert Fig 4 here

The majority of the male participants aged between 41-50 years agreed regarding the lack of knowledge and confidence, as well as lack of time and interest as a barrier (p<0.05). Most participants between 31-40 years age group or having 16-20 years of job experience agreed that legal liability can be a barrier in reporting an ADR (p<0.05). The majority of participants aged between 31-35 years agreed that increased reminders from the National PV Centre (NPC), the establishment of an online system, continuous medical education, training, and educational seminars could improve ADR reporting (p<0.05) as shown in Fig 5. “

Q. I'm not sure the IQR / median add anything given the use of a 3 point scale. Tables would be clearer without them.

Response: Many thanks, we have removed it as suggested.

Q. Figure 4 (line 490) is labelled as Figure 2

Response: Thank you, its corrected.Line 571

Q. Figure 3 and 4 take different styles despite presenting equivalent data. If you plan to retain figures please consider consistency of style.

Response: Figures were changed to uniform style.

Q. Tables 3 and 4 I found quite hard to pull out the key data. It would be clearer to present just the significant differences and add a statement to say all others were non-sig. Explain 'category' means type of HCP. consider if you can include the direction of difference within the table rather than having to jump back and forth from table to text to see where the difference lay.

Response: Thank you, we have edited the text as suggested.

Q. Please review the discussion to ensure all studies mentioned are referenced e.g. lines 280-282, 286

Response: This is being done now in line 287-292

Q. 250-51 you say that providing education would increase active participation - studies have shown this link so please provide evidence in the form of some reference(s).

Response: Many thanks for pointing it out. Relevant references have been added and they show that educating healthcare professionals can increase active participation. Line 243-247

Q. Time is a major barrier reporting in most studies of this type - I found the section 252-265 to be confusing as it took a long time to work out that this was providing an explanation for the barrier of time - make it more explicit upfront but also raise the fact that this comes up almost universally in studies re barriers to reporting (i.e, this is not unique or only related to the issue of workforce moving abroad).

Response: Thank you, we have rephrased the text to make the point clearer to the audience.

Lack of time to report an ADR was highlighted as an important barrier by the majority of HCPs. This could be due to overly occupied staff in hospitals and workforce shortages. There were 195,896 physicians, 99228 nurses, and 32511 pharmacists in Pakistan’s public sector hospitals (9). According to an estimate, the population to health services ratio for physicians is 0.82 (while the standard physician to population ratio is 1:1000), for nurses and midwives 0.57 per 1000 population (the standard ratio is 3 nurses per 1000), and 0.9 pharmacists per 100,000 population (whereas the WHO recommended ratio is 1 pharmacist per 2000 population) (38, 39). This low ratio of HCPs to patients could result in increased workload for the rest of HCPs, ultimately leading to the lack of time to report an ADR. The workforce shortages are also illustrated with the fact that every year the majority of HCPs including physicians, pharmacists, and nurses move overseas to look for better opportunities. According to an estimate, around 17.6% of physicians and 15% of nurses migrated to developed countries for better working opportunities, which causes the shortage of skilled healthcare workers leading to the extra workload on the existing HCPs. (40). Improvement can be made if the government pays proper attention to the sufficient staffing of HCPs in the hospitals. This could not only decrease the time for the reporting of a suspected ADR but also could help reduce the chance of making errors during the medicines use process (41). 

Q. 266-272 you discuss confidence yet confidence in your study seemed lower than in other papers - why might this be?

Response: This could be due to the weakness of health system, as healthcare professionals seem to have less trust and confidence in sharing ADR data and information. This could be manifold including whether this sharing would be any useful to improve the quality of care for the patients. Line 265-274

Q. 276-279 interesting re the app - is there any evidence yet that it has improved reporting? If not then just add that it's too early to see the results.

Response: Yes, added as suggested to line 284-286

Q. 280-282 ADR reporting is NOT mandatory in the UK for physicians, although strongly encouraged (and I don't believe this is the case in some of the other countries) - where did you get this information from?

Response: Many thanks for pointing this out. Yes, it varies from country to country, there was some information missed in this regard. We have updated the information. Line 287-289

Q. Please add some critical commentary around limitations. The conclusion should be a bit more focussed to make clear the context and any limitations which should be borne in mind while considering these conclusions.

Response: Thank you, we have added the section. Line 320-336

“The study has some limitations, such as, we used self-reported responses as the method of inquiry, which could have led to the recall and social desirability bias. However, as the self-administered questionnaire did not include participants’ names and data were anonymously recorded, thus the potential of social desirability was reduced. 

The cross-sectional study design selection may have led to many gaps, which could be otherwise filled if used a time series analysis to explain the factors responsible for ADR underreporting. Additionally, there was an over-representation of nurses and physicians over pharmacists, owing to the current healthcare team dynamics in the Pakistani healthcare setting, whereby pharmacists are still under-recruited in public hospitals. 

The findings of the study cannot be generalized to the whole country as this study was conducted among HCPs from Lahore, Punjab. However, the study was conducted in the second-largest city, Lahore with more advanced and facility-based systems. Thus, despite these limitations, the information is still useful for both HCPs as well as for policymakers.

A key strength of this study was the inclusion of physicians, pharmacists, and nurses working in tertiary care public hospitals. This helped to identify the range of challenges encountered in most facility-based healthcare settings. Being the study is important as it has generated key insights on the barriers and facilitators to improve the overall PV system in the country.”

---

## [Decision Letter · Decision Letter 1]

17 May 2022

PONE-D-21-35762R1Barriers and facilitators to Pharmacovigilance activities in Pakistan: A healthcare professionals-based surveyPLOS ONE

Dear Dr. Hussain,

Thank you for submitting your manuscript to PLOS ONE. After careful consideration, we feel that it has merit but does not fully meet PLOS ONE’s publication criteria as it currently stands. Therefore, we invite you to submit a revised version of the manuscript that addresses the points raised during the review process.

We look forward to receiving your revised manuscript.

Kind regards,

Vijayaprakash Suppiah, PhD

Academic Editor

PLOS ONE

Journal Requirements:

Reviewers' comments:

Reviewer's Responses to Questions

**Comments to the Author**

1. If the authors have adequately addressed your comments raised in a previous round of review and you feel that this manuscript is now acceptable for publication, you may indicate that here to bypass the “Comments to the Author” section, enter your conflict of interest statement in the “Confidential to Editor” section, and submit your "Accept" recommendation.

Reviewer #1: (No Response)

2. Is the manuscript technically sound, and do the data support the conclusions?

Reviewer #1: Yes

3. Has the statistical analysis been performed appropriately and rigorously? 

Reviewer #1: Yes

4. Have the authors made all data underlying the findings in their manuscript fully available?

Reviewer #1: Yes

5. Is the manuscript presented in an intelligible fashion and written in standard English?

Reviewer #1: No

6. Review Comments to the Author

Reviewer #1: The authors have addressed the previous comments. However, the comments below need to be addressed in the revised version of the manuscript. Additionally, as this article requires major language editing, it need to be proofread by a native English speaker.

Abstract

---------

1. Page 2, line 34-35: "However, barriers to the underreporting of ADRs exist in almost all healthcare systems." This statement does not sound quite right. Barriers exist causing underreporting of ADR rather than barriers to underreporting of ADR. Same in line 44-45.

Introduction

--------------

1. Page 3, Line 61-62: Add reference for the statement "Among hospitalized patients, ADRs represent the fifth most common cause of death."

2. Page 4, Line 82-83: Restructure the sentence so that the 's does not appear after the abbreviation.

3. Page 4, Line Line 96-97: Repeated information.

4. Page 5, Line 115: Repeated use of the abbreviation PV "study has analyzed the core indicators responsible for a functional PV system (PV) in Pakistan." Add reference to this statement as well.

5. Page 5, Line 119:What is the difference between the barriers and facilitators affecting the ADR reporting system and barriers and facilitators to improve the underreporting of ADRs?

Methods

---------

1. Page 6, Line 114: Repeated use of the abbreviation HCP.

7. PLOS authors have the option to publish the peer review history of their article (what does this mean?). If published, this will include your full peer review and any attached files.

Reviewer #1: No

---

## [Author Response · Author response to Decision Letter 1]

19 May 2022

Dear Reviewer, we would like to thank you for reviewing the manuscript and helping us to improve the quality of our manuscript.

As suggested, the manuscript has been proofread and edited by a native English speaker.

Abstract

---------

1. Page 2, line 34-35: "However, barriers to the underreporting of ADRs exist in almost all healthcare systems." This statement does not sound quite right. Barriers exist causing underreporting of ADR rather than barriers to underreporting of ADR. Same in line 44-45.

Response: Many thanks, we have modified the sentence as suggested.

Line 34-35: However, in almost all healthcare systems barriers exist that lead to the underreporting of ADRs.

Line 44-45: …………………. lack of support from a colleague as a reason that could lead to the underreporting of ADR.

Introduction

--------------

1. Page 3, Line 61-62: Add reference for the statement "Among hospitalized patients, ADRs represent the fifth most common cause of death."

Response: Thank you. The reference has been added.

Line 63: European Commission Strengthening Pharmacovigilance to Reduce Adverse Effects of Medicines. Memo/08/782. 2008. [(accessed on 18 May 2022)]. pp. 2–5. Available online: https://ec.europa.eu/commission/presscorner/detail/de/MEMO_08_782.

2. Page 4, Line 82-83: Restructure the sentence so that the 's does not appear after the abbreviation.

Response: Many thanks, we have corrected the sentence.

Line 84-85: Pakistan joined the PIDM by the World Health Organization (WHO) as the 134th member in 2018.

3. Page 4, Line 96-97: Repeated information.

Response: Yes, repeated information has now been removed. Line 98

4. Page 5, Line 115: Repeated use of the abbreviation PV "study has analyzed the core indicators responsible for a functional PV system (PV) in Pakistan." Add reference to this statement as well.

Response: Thank you, we have removed the repeated word “PV” from the sentence and have added the reference.

Line 116: A recent study has analyzed the core indicators responsible for a functional pharmacovigilance system in Pakistan.

Reference: 19. Khan M, Hamid S, Ur-Rehman T, Babar ZJFPdf. Assessment of the Current State of PV System in Pakistan Using Indicator-Based Assessment Tool. 2022.

5. Page 5, Line 119: What is the difference between the barriers and facilitators affecting the ADR reporting system and barriers and facilitators to improve the underreporting of ADRs?

Response: Many thanks for pointing it out. We have edited the sentence for clarity.

Line 119: there is a scarcity of literature about barriers causing underreporting of ADRs and facilitators to improve the reporting of ADRs among healthcare professionals in the country.

Methods

---------

1. Page 6, Line 114: Repeated use of the abbreviation HCP.

Response: Thanks, we have removed the repeated word HCP.

Line 145: The questions included demographic details and experiences of HCPs regarding barriers and facilitators toward ADR reporting in their work setting.

---

## [Decision Letter · Decision Letter 2]

16 Jun 2022

PONE-D-21-35762R2Barriers and facilitators to pharmacovigilance activities in Pakistan: A healthcare professionals-based surveyPLOS ONE

Dear Dr. Hussain,

Thank you for submitting your manuscript to PLOS ONE. After careful consideration, we feel that it has merit but does not fully meet PLOS ONE’s publication criteria as it currently stands. Therefore, we invite you to submit a revised version of the manuscript that addresses the points raised during the review process.

We look forward to receiving your revised manuscript.

Kind regards,

Vijayaprakash Suppiah, PhD

Academic Editor

PLOS ONE

Journal Requirements:

Reviewers' comments:

Reviewer's Responses to Questions

**Comments to the Author**

1. If the authors have adequately addressed your comments raised in a previous round of review and you feel that this manuscript is now acceptable for publication, you may indicate that here to bypass the “Comments to the Author” section, enter your conflict of interest statement in the “Confidential to Editor” section, and submit your "Accept" recommendation.

Reviewer #1: (No Response)

2. Is the manuscript technically sound, and do the data support the conclusions?

Reviewer #1: Yes

3. Has the statistical analysis been performed appropriately and rigorously? 

Reviewer #1: Yes

4. Have the authors made all data underlying the findings in their manuscript fully available?

Reviewer #1: Yes

5. Is the manuscript presented in an intelligible fashion and written in standard English?

Reviewer #1: Yes

6. Review Comments to the Author

Reviewer #1: Thank you for the opportunity to review the revised manuscript entitled “Barriers and facilitators to pharmacovigilance activities in Pakistan: A healthcare professionals-based survey”. The authors have conducted a questionnaire-based study to determine the barriers and facilitators related to adverse drug reaction (ADR) reporting of physicians, pharmacists and nurses in Lahore, Pakistan. Overall, the authors have addressed the comments raised in a previous round of review. However, there are some corrections that still need to be made. Generally, the use of abbreviation in this manuscript is extensive but unstandardized and inconsistent. I urge that the authors pay attention to the correct use of abbreviations in the manuscript and strive for consistency. Other suggested corrections are as listed below.

Abstract

----------

1. Page 2, Line 41: “Over sixty-two percent of physicians…” change to “Over 62% of physicians…”

2. Page 2, Line 43: “Similarly, 41.6% of 44 nurses identified a lack of financial reimbursement…” similar to? Unclear use of adverb.

Introduction

---------------

1. Page 4, line 95: Abbreviate Drug Regulatory Authority of Pakistan as (DRAP) at first mention.

2. Page 5, line 106: Already abbreviated in line 88. Since it is the beginning of the sentence, the full form can be used without stating the abbreviation.

3. Page 5, line 106: Healthcare professionals play an important role in the establishment and “working” of pharmacovigilance in a country. Suggest to replace “working” with “functionality”.

4. Page 5, line 107: Hence it is important to understand “the” barriers and facilitators…

Results

--------

1. Page 8, Line 191: A rate has to be %, not a number. Suggest to rephrase to “The returned questionnaires were collected by the researcher with a response rate of 90.1% (n=346) for all participating HCPs.”

2. Page 9, Line 209: spacing between “and” and “17”.

Discussion

------------

1. Page 10, Line 248: This “could be” evident from the studies… Change “could be” to “was”.

7. PLOS authors have the option to publish the peer review history of their article (what does this mean?). If published, this will include your full peer review and any attached files.

Reviewer #1: No

---

## [Author Response · Author response to Decision Letter 2]

17 Jun 2022

Dear Reviewer, we would like to thank you for reviewing the manuscript and helping us to further improve the quality of our manuscript.

Abstract

1. Page 2, Line 41: “Over sixty-two percent of physicians…” change to “Over 62% of physicians…”

Response: Thank you, the suggestion is accepted. Line 41

2. Page 2, Line 43: “Similarly, 41.6% of 44 nurses identified a lack of financial reimbursement…” similar to? Unclear use of adverb.

Response: Many thanks, we have replaced “similarly” with “furthermore”. Line 44

Introduction

1. Page 4, line 95: Abbreviate Drug Regulatory Authority of Pakistan as (DRAP) at first mention.

Response: Noted and abbreviations are made standardized throughout the manuscript. Line 96

2. Page 5, line 106: Already abbreviated in line 88. Since it is the beginning of the sentence, the full form can be used without stating the abbreviation.

Response: Many thanks and we have made the change as suggested. Line 107

3. Page 5, line 106: Healthcare professionals play an important role in the establishment and “working” of pharmacovigilance in a country. Suggest to replace “working” with “functionality”.

Response: Thank you and we have replaced the word as suggested. Line 108

4. Page 5, line 107: Hence it is important to understand “the” barriers and facilitators…

Response: Thank you for pointing it out. We have added “the” as suggested. Line 108

Results

1. Page 8, Line 191: A rate has to be %, not a number. Suggest to rephrase to “The returned questionnaires were collected by the researcher with a response rate of 90.1% (n=346) for all participating HCPs.”

Response: Many thanks. We have rephrased the sentence as advised. Line 192

2. Page 9, Line 209: spacing between “and” and “17”.

Response: Thank you, space is provided between “and” and “17”. Line 210

Discussion

1. Page 10, Line 248: This “could be” evident from the studies… Change “could be” to “was”.

Response: Thank you, we have replaced “could be” with “was”. Line 249

---

## [Decision Letter · Decision Letter 3]

4 Jul 2022

Barriers and facilitators to pharmacovigilance activities in Pakistan: A healthcare professionals-based survey

PONE-D-21-35762R3

Dear Dr. Hussain,

We’re pleased to inform you that your manuscript has been judged scientifically suitable for publication and will be formally accepted for publication once it meets all outstanding technical requirements.

Kind regards,

Vijayaprakash Suppiah, PhD

Academic Editor

PLOS ONE

Reviewers' comments:

Reviewer's Responses to Questions

**Comments to the Author**

1. If the authors have adequately addressed your comments raised in a previous round of review and you feel that this manuscript is now acceptable for publication, you may indicate that here to bypass the “Comments to the Author” section, enter your conflict of interest statement in the “Confidential to Editor” section, and submit your "Accept" recommendation.

Reviewer #1: All comments have been addressed

2. Is the manuscript technically sound, and do the data support the conclusions?

Reviewer #1: Yes

3. Has the statistical analysis been performed appropriately and rigorously? 

Reviewer #1: Yes

4. Have the authors made all data underlying the findings in their manuscript fully available?

Reviewer #1: (No Response)

5. Is the manuscript presented in an intelligible fashion and written in standard English?

Reviewer #1: Yes

6. Review Comments to the Author

Reviewer #1: (No Response)

7. PLOS authors have the option to publish the peer review history of their article (what does this mean?). If published, this will include your full peer review and any attached files.

Reviewer #1: No

---

## [Editor Report · Acceptance letter]

11 Jul 2022

PONE-D-21-35762R3 

Barriers and facilitators to pharmacovigilance activities in Pakistan: A healthcare professionals-based survey 

Dear Dr. Hussain:

I'm pleased to inform you that your manuscript has been deemed suitable for publication in PLOS ONE. Congratulations! Your manuscript is now with our production department. 

Kind regards, 

on behalf of

Dr. Vijayaprakash Suppiah 

Academic Editor

PLOS ONE